# From Spaceflight to Mars *g*-Levels: Adaptive Response of *A. Thaliana* Seedlings in a Reduced Gravity Environment Is Enhanced by Red-Light Photostimulation

**DOI:** 10.3390/ijms22020899

**Published:** 2021-01-18

**Authors:** Alicia Villacampa, Malgorzata Ciska, Aránzazu Manzano, Joshua P. Vandenbrink, John Z. Kiss, Raúl Herranz, F. Javier Medina

**Affiliations:** 1Centro de Investigaciones Biológicas Margarita Salas (CSIC), Ramiro de Maeztu 9, 28040 Madrid, Spain; avillacampa@cib.csic.es (A.V.); mciska@cib.csic.es (M.C.); aranzazu@cib.csic.es (A.M.); 2School of Biological Sciences, Louisiana Tech University, Ruston, LA 71272, USA; jpvdb@latech.edu; 3Department of Biology, University of North Carolina-Greensboro, Greensboro, NC 27402, USA; jzkiss@uncg.edu

**Keywords:** microgravity, partial gravity, transcription factors, gene expression, root meristem

## Abstract

The response of plants to the spaceflight environment and microgravity is still not well understood, although research has increased in this area. Even less is known about plants’ response to partial or reduced gravity levels. In the absence of the directional cues provided by the gravity vector, the plant is especially perceptive to other cues such as light. Here, we investigate the response of *Arabidopsis thaliana* 6-day-old seedlings to microgravity and the Mars partial gravity level during spaceflight, as well as the effects of red-light photostimulation by determining meristematic cell growth and proliferation. These experiments involve microscopic techniques together with transcriptomic studies. We demonstrate that microgravity and partial gravity trigger differential responses. The microgravity environment activates hormonal routes responsible for proliferation/growth and upregulates plastid/mitochondrial-encoded transcripts, even in the dark. In contrast, the Mars gravity level inhibits these routes and activates responses to stress factors to restore cell growth parameters only when red photostimulation is provided. This response is accompanied by upregulation of numerous transcription factors such as the environmental acclimation-related WRKY-domain family. In the long term, these discoveries can be applied in the design of bioregenerative life support systems and space farming.

## 1. Introduction

The achievement of plant cultivation in space, also called “space farming,” is an important step in the development of bioregenerative life support systems to enable long-term space exploration, since plants are fundamental elements for oxygen and nutrient supplies as well as waste recycling [1]. With this objective, it is important to study the response of plants to the space environment. Plants have been successfully grown in space on numerous occasions [2], even though major physiological changes, such as the alteration of cell proliferation rate and ribosome biogenesis, have been reported [3]. Most major physiological changes are regulated and tuned by phytohormones and transcription factors (TFs). These latter function as molecular switches activating or repressing the expression of genes or sets of genes in response to different stimuli, e.g., changes in the environmental conditions. Some changes in the phytohormone levels have been previously reported in experiments performed in real and simulated microgravity, such as a different distribution of cytokinin in real microgravity [4] and auxin accumulation in simulated microgravity [5]. However, more attention has been given to the changes in plant physiology (e.g., response to hypoxia, cell wall modifications, accelerated cell cycle) rather than to the hormone regulatory pathways and TFs.

Space exploration involves the exposure of plants to microgravity conditions, as they exist on spacecraft and stations orbiting the Earth, such as the International Space Station (ISS). Although microgravity effects have been extensively studied in living organisms, they are difficult to overcome since plants, like any other terrestrial organisms, have evolved in a constant gravity vector. Plants orient their growth according to the gravity vector (gravitropism), with positive root gravitropism and negative shoot gravitropism. Nevertheless, in microgravity, the cue for this tropism (i.e., the gravity vector) is not present.

Other tropisms are also involved in directing plant growth. For instance, using light as the tropistic cue, phototropism drives plant growth orientation with a negative root phototropism and positive shoot phototropism [6]. The interaction among gravitropism, phototropism and other tropisms, such as hydrotropism [7] and thigmotropism [8], produces the overall direction of plant growth [9], which is constantly adapted to the changing environmental conditions. These well-established positive and negative tropisms on Earth must be reevaluated in space since, in the absence of the gravity vector, new phototropic responses can be observed that were masked by gravitropism in the Earth. In fact, root positive phototropic response to red light [10] and blue light [11] have been reported in spaceflight studies. Different wavelengths of light are known to promote different responses on plant growth and development [12,13]. Thus, specific light conditions could be applied to overcome some of the deleterious effects on plant growth and development induced by microgravity. For example, red light is known to stimulate cell proliferation and promote ribosome biogenesis [14], both processes affected by spaceflight. In fact, red light has already been used in a similar experiment in simulated microgravity and was applied to *Arabidopsis thaliana* seedlings as a part of the Seedling Growth (SG) series (SG1 and SG2 experiments on the ISS) [15]. The results obtained were positive, in the sense of compensating at least a part of the alterations induced by microgravity.

Furthermore, the influence of partial or reduced gravity levels on the plant physiology should be investigated to enable human settlements on nearby planets [16]. In recent years, special attention has been given to Mars. Little is known so far on the plant response to partial gravity levels, which is important considering space agencies’ plans to travel back to the Moon (Deep Space Gateway, DSG) in 2024 [17] and to Mars in the near future. With the purpose of studying how partial gravity levels can affect plant development, some studies have used analogs, such as random positioning machines (RPMs), to reproduce Moon or Mars gravity levels and study their effect on Earth [18]. Simultaneously, the European Modular Cultivation System (EMCS, [19]), which was installed in the ISS from 2008 to 2018, provided the ability to apply different *g*-forces in space by means of a built-in centrifuge. The SG experiment was executed in this hardware to test the contribution of red and blue light stimulation interaction with the reduced gravity stimuli [20]. Firstly, we used the EMCS to investigate transcriptomic changes in *A. thaliana* seedlings exposed to different *g*-levels for the last two days with blue light stimulation in the SG series (SG1 and SG2). We applied different gravity levels (microgravity, 0.1*g*; Moon; Mars; near earth *g*-level; 1*g*) to blue-light stimulated wild-type (WT) Landsberg ecotype *A. thaliana* seedlings and demonstrated a replacement of gravitropism by blue-light-based phototropism signaling at microgravity level [21], but a striking stress response was found at 0.1*g*. We also determined different components of the transcriptional response to the lack of gravity as the *g*-gradient is progressively reduced [22].

The use of transcriptomic techniques has provided vast data on gene expression in plants grown in space. *A. thaliana* is so far the most widely studied plant in space biology using omics techniques [21,22,23,24,25,26,27,28,29,30,31] and microscopic methods [4,32], although a few crop species have been recently incorporated to space studies [32,33,34,35,36]. Scarce material, high cost, and extensive logistics are highly limiting factors for space experiments, making the investigation of plant response to the space environment challenging, reinforcing the requirement of better controls and complementary research in ground simulation and reference facilities [37]. Moreover, there is a growing awareness in the space biology community to define and use the same criteria when describing the spaceflight experiment metadata so the cross-comparisons between the spaceflight experiments can be performed more rigorously [38,39].

Here, we combined morphological and molecular approaches to describe the changes in 6-day-old *A. thaliana* seedlings (Col-0) grown in the SG experiments (SG2 and SG3) in three *g*-levels (microgravity, Mars gravity level and 1*g* ground reference run (GRR)) under red light photostimulation, and a control in darkness for the last two days of the experiment. In addition, we compared our results to other transcriptomic data obtained from *A. thaliana* seedlings grown during spaceflight and available in the GeneLab database ([40]; https://genelab.nasa.gov/) for further validation of our results. In the long term, our studies will pave the way to understand the molecular mechanisms to improve the cultivation conditions of plants on other planets.

## 2. Results

### 2.1. Anatomic Changes in Microgravity and Partial Gravity in Different Light Conditions

We analyzed the morphology of *A. thaliana* seedlings grown on the EMCS in the ISS. In our experimental conditions, the seeds germinated in altered gravity levels giving us a chance to observe how the plant deals with the new environment, meaning how they acclimate. The germination rate during the space experiment was similar to the one in the GRR (ISS samples 96.7%; GRR 93.93%) suggesting the plant activates these acclimation mechanisms already during germination.

We investigated the influence of different gravity levels and light conditions on meristem organization and size expressed in the length of the meristem (the distance from quiescent center to the first elongated cell in the epidermis) and in the number of meristematic cells in the epidermis (Figure 1A). The typical organization of the meristem with easily distinguished quiescent center and three layers of meristematic cells (epidermis, endodermis and cortex) was observed in all the conditions. These results suggest that altered gravity levels, and in general spaceflight conditions, do not disturb the well-conserved organization of the meristem in *A. thaliana*. The root cap columella also displayed its typical organization with the first meristematic layer followed by three to four layers of gravity-perceiving statocytes [41]. No difference in the number of layers of columella cells was observed among the conditions. In respect to the meristem length, in the seedlings grown in the dark, no significant changes were observed at any *g*-level. However, in the seedlings photostimulated with red light, a gradual increase in the meristem size was observed with the decrease of *g*-level (1*g* GRR < Mars < µ*g*), although the difference between Mars and µg was not significant (Figure 1B). In addition, the length of the meristem and the number of meristematic cells per meristematic layer were increased in the photostimulated seedlings in comparison to the seedlings grown at the same *g*-level in darkness, although the difference was only statistically significant at the Mars gravity level (Figure 1B). These observations were similar to those published in previous reports showing that red light stimulated proliferation [14,15].

Next, we estimated the nucleolar activity by measuring the area of immunofluorescent staining using an antibody against the nucleolar protein fibrillarin, in different *g*-levels and light conditions (Figure 1C,D). Fibrillarin is a well-known and abundant nucleolar protein involved in pre-rRNA processing regulation [42], and it can be used as a nucleolar marker. Under standard conditions of growth, the size of the nucleolus in the meristematic cells is directly related to its activity, determined by the production of the ribosomal units [43]. Therefore, the size and the structural features of the nucleolus are a reliable marker of the rate of ribosome biogenesis [44], which is determined by the demand in protein synthesis, meaning the higher the nucleolar size the higher protein production. A reduction in the size of the nucleolus was observed in meristems of the seedlings grown at Mars *g*-level without photostimulation. This reduction was significant in comparison to the nucleolus in red-light photostimulated seedlings at the same gravity level, and in comparison to the seedlings grown in 1*g* GRR and in microgravity without photostimulation. This indicates that the protein biosynthesis was also reduced in this condition. Red light seems to have a positive effect on the nucleolar activity at the Mars *g*-level, since meristematic nucleoli in red photostimulated seedlings at this gravity level display similar size to the GRR seedlings. Surprisingly, nucleoli in seedlings grown in microgravity in both light conditions also had similar size as nucleoli in GRR seedlings (Figure 1D). 

To investigate in more detail the changes that nucleolus undergoes in different light and gravity conditions, we analyzed nucleolar ultrastructure using TEM. In most conditions, nucleoli had a regular round shape and a typical structure, where three components could be distinguished: granular component (GC), dense fibrillar component (DFC) and fibrillar centers (FC) (Figure 1E). Despite the fact that we have not observed in microgravity conditions without photostimulation the reduction in nucleolar size that was observed before in etiolated plants in the ROOT experiment [3], we could confirm in TEM images that nucleoli presented features typical for low-active nucleoli (low content of GC and heterogeneous FCs with condensed chromatin inside) [45]. However, in nucleoli of red-light photostimulated seedlings exposed to microgravity and partial gravity, features of active nucleoli, such as abundant GC intermingled with DFC and small FCs, were observed. No condensed intranucleolar chromatin was observed in these samples. This observation agrees with previous reports from our ROOT spaceflight experiment, where the combined effect of etiolation and microgravity caused a significant reduction in nucleolar activity [3]. Since, in the SG2 and SG3 experiments, the seedlings germinated and grew for four days with a photoperiod, the inhibitory effect of microgravity and darkness (last two days of culture) on nucleolar activity and nucleolar size could be diminished with respect to the experiments using etiolated seedlings.

### 2.2. Global Transcriptomics 

Principal component analysis (PCA) of the replicates is shown in Figure 2A. The clustering of the replicates is consistent. There is a very clear separation among the three *g*-levels in the samples exposed to red-light photostimulation (µgrl, Marsrl and grrrl) but it is not well distinguished between the Marsd and grrd samples. 

To elaborate on this, we applied two different approaches to perform the comparisons in the transcriptomics data. First, to investigate the effect of different gravity levels on *A. thaliana* seedlings, we compared the transcriptomes of samples grown in microgravity or Mars gravity level to the same light condition in 1-*g* GRR transcriptome as the reference; µgd-grrd, Marsd-grrd, µgrl-grrrl; Marsrl-grrrl. Next, to dissect the effect of red light at different *g*-levels, we compared the transcriptomes of seedlings grown in the same *g*-levels but in different light conditions; µgrl-µgd; Marsrl-Marsd, grrrl-grrd. In both cases, we used a q-value < 0.05 and a threshold fold change of Log_2_FC ± 1.5.

In the gravity level comparisons, more than 600 differentially expressed genes (DEGs) were upregulated in µgd, µgrl, and Marsrl, whereas in Marsd only half of this number (372) (Figure 2B). These results show that transcriptomes of the seedlings photostimulated by red light presented a similar number of upregulated DEGs at both gravity levels, but in seedlings grown in dark, twice as many DEGs were upregulated in microgravity than in Mars gravity. Approximately half of the upregulated DEGs are common for each gravity level independently of being exposed to darkness or photostimulation (253 and 175 in microgravity and Mars, respectively). Gene Ontology (GO) analysis of biological process categories of those sub-lists are shown in Appendix A. The low number of upregulated genes in the Marsd sample is in accordance with a small size of meristematic nucleoli in this condition, suggesting a reduced rate of protein biosynthesis, meaning that the transcriptome status is reflected at the proteome level. This effect could be a result of the upregulation of Ovate Family Protein 10 (OFP10), a transcription repressor [46] upregulated only in Marsd condition (Log_2_FC 2; all Log_2_FC values given in the text are statistically significant; q-value < 0.05). 

Only 15 DEGs are upregulated in all four comparisons. Around 500 genes were downregulated in µgd, Marsd, and Marsrl, whereas in µgrl, there were twice as many (1012 genes). In this case, we observed twice the number of downregulated genes in the red-light photostimulated seedlings in microgravity in comparison to Mars *g-*level. This number is particularly high (615 DEGs) in µgrl only, and specifically enriched in photosynthesis function (Appendix A). A total of 55 DEGs are downregulated in the four conditions. There are also nearly a hundred up- and down-regulated genes in the red-light conditions (µgrl and Marsrl), which include abiotic stress responses in the upregulated DEG and metabolic biosynthetic pathways in the downregulated genes (Appendix A). In summary, seedlings grown at Mars *g-*level in darkness seem to be the least altered samples (in agreement with the PCA), while the ones grown in microgravity and photostimulated with red light show a high number of DEGs. Given the small variations observed in the plant anatomy, as reported in the preceding section, this dysregulation does not necessarily mean an adverse effect on the plant. Most likely, the changes in transcript levels also involve genes associated with the acclimation that the seedlings experience from the germination and during six-day exposure to altered gravity level. The red-light photostimulation comparison is discussed below.

In the Gene Ontology (GO) analysis, we used the total number of upregulated or downregulated DEGs to look for common and specific altered molecular functions in each condition. We observed that in the upregulated genes there was a clear clustering of common categories by *g-*levels rather than by light conditions (Figure 2C, left). Among the categories upregulated in all conditions, response to osmotic stress, wounding and cellular response to hypoxia could be identified. Photosynthesis categories are highly upregulated in microgravity, but not at Mars *g-*level. It is surprising that this category is equally upregulated in both light conditions, the red-light photostimulated sample and seedlings grown in darkness for the last two days. Another strongly upregulated category in microgravity was oxidative phosphorylation. On the other hand, categories specific for Mars conditions (both light treatments) include response to ethylene, drug, defense response and positive regulation of biosynthetic processes and organ growth.

Among the downregulated DEGs (Figure 2C, right), the common functions are related to hypoxia, while the rest of the downregulated functions seem to be specific for each *g-*level and light condition. In addition, in the µgrl sample, photosynthesis and light harvesting category was strongly downregulated. This result is in agreement with the previous results obtained in the SG1–2 experiments in the Ler ecotype, using blue-light stimulated seedlings [21], confirming the role of light and phototropism as an alternative cue for plant development in the total absence of gravity. To determine if the same genes are being downregulated, we compared the two datasets and found 16 common genes. Even if the overlap of downregulated genes was not striking, there was an enrichment of photosynthesis function in those 16 genes specifically related to light harvesting in photosystem I and II, and five downregulated light-harvesting chlorophyll *a*/*b* binding (LHCB) proteins (Appendix A). These results suggest that the same function is affected by microgravity in both spaceflight experiments (i.e., one with red-light stimulation and the other with blue light). 

Extended heatmaps with top 100 enriched clusters are shown in Appendix A. Common GO categories are either upregulated in both gravity conditions, such as the response to water, response to reactive oxygen species, response to osmotic stress, response to wounding and cellular response to hypoxia, or downregulated, such as cellular response to decreased oxygen levels. However, only a few or none of the dysregulated transcripts are common for both conditions in each category (Appendix A).

### 2.3. Dysregulation of Transcriptional Factors (TFs) and Hormonal Pathways in Microgravity and Partial Gravity

Many of the functional categories that are dysregulated in both microgravity and Mars gravity conditions such as osmotic and biotic stresses or response to hypoxia are regulated by both phytohormones and families of transcription factors [47,48,49,50]. Among the most over-represented TF families in the DEGs, we encountered the WRKY domain family, which forms one of the largest TF families in flowering plants, as well as other large families of TFs such as ethylene responsive factor (ERF), ATAF1/2 CUC2 (cup-shaped cotyledon) (NAC) and myeloblastosis (MYB). We tested by Chi-squared analyses whether a g iven TF family is overrepresented in each of the conditions and discovered that WRKY and NAC TFs are overrepresented in Mars *g*-level (both light conditions), and MYB TFs are overrepresented in Mars *g-*level (both light conditions) and in microgravity red-light photostimulated samples. In contrast, ERFs are overrepresented in all four conditions. Since WRKY TFs have an important role in plant acclimation and are “multifunctional switches,” we focused on this family of TFs (reviewed in [47]). Data on WRKYs upregulated in the Mars conditions (both lights) are presented in Table 1. Although in microgravity WRKY were not overrepresented, one family member, *At*WRKY63, was significantly upregulated in both light conditions (µgd-grrd Log_2_FC 2.56 and µgrl-grrrl Log_2_FC 2.79).

We evaluated the influence of microgravity and Mars *g-*level on hormonal pathways using Kyoto Encyclopedia of Genes and Genomes (KEGG) pathway analyses. Few important hormonal pathways were significantly affected by microgravity and partial gravity, as seen in Figure 3 for the “plant hormone signal transduction pathway” (ath04075), including the expression level and significance for the four light/gravity conditions under study (µgd-grrd, µgrl-grrrl, Marsd-grrd, Marsrl-grrrl). Each signaling step includes the information of transcription levels of one or more genes involved in signal transduction (full list of genes is available at KEGG database under the ath04075 pathway identifier). According to KEGG Pathway analysis, the auxin pathway was activated at different steps in microgravity conditions (GRETCHEN HAGEN 3 (GH3) step in µgd and small auxin upregulated RNA (SAUR) in µgrl and repressed in Marsrl condition (GH3 step), suggesting cell enlargement and plant growth are promoted in microgravity but not at Mars *g-*level. *GH3* genes, downregulated in µgd (*DFL1*) and upregulated in Marsrl (*GH3.3* and *AT1G48660*) encode auxin-amido synthetases and promote the inactivation of indole acetic acid (IAA) [51]. Eleven SAUR genes which regulate auxin-mediated growth are upregulated in µgrl sample.

The cytokinin pathway was significantly activated (Log_2_FC > 1.5) in microgravity through histidine phosphotransfer proteins (AHPs), which function as positive regulators of cytokinin signaling [52]. From the six members of this gene family expressed in *A. thaliana,* two were upregulated in microgravity: *AHP3* and *AHP4*. In Marsd conditions, this route was significantly inhibited at the CRE1 step (histidine kinase 2, *HK2*). The gibberellin pathway was significantly repressed at the TF step in microgravity (*PIF4, PIL6*). 

At least one step of the abscisic acid (ABA) pathway is clearly activated in all conditions except for µgd (Figure 3). Among the upregulated genes that contribute to this activation are highly ABA-induced *PP2C* gene 2 (*HAI2*) and *HAI3* proteins and, in the case of the Marsrl samples, also SNF1-related protein kinase 2.9 (*SNRK2.9*) and *SNRK2.5.* The ethylene pathway was activated in all conditions; at EIN3 step in microgravity (*AT5G65100*) and ERF1/2 at Mars *g*-level (*ERF1, ERF2*), which is in agreement with the GO analysis and the overrepresentation of ERFs in the upregulated genes in all conditions. 

The brassinosteroid pathway is significantly inhibited through BRI1 suppressor 1 (BSU1) protein in µgrl and Mars samples. In addition, *TOUCH 4 (TCH4*), which is involved in the response to mechanical stimulus, cold, hypoxia and regulates cell elongation (arabidopsis.org), is downregulated in all light and gravity conditions (although with threshold Log_2_FC > 1.5 only in µgrl sample). Both microgravity and partial gravity influenced the transcription of jasmonate-ZIM domain (JAZ) proteins, which are transcriptional repressors in jasmonic acid (JA) signaling. *JAZ1* was upregulated in Mars gravity and *JAZ4* in microgravity. Additionally, *ABA-inducible BHLH-type transcription factor* (*AIB*) that interconnects JA-ABA pathways was also downregulated in microgravity conditions. The salicylic acid pathway was repressed in microgravity (*PR1-like*, *AT1G50060*) and activated in the Marsrl sample (*AT4G33720*) at the PR-1 step. 

Similar analysis with light comparisons (µgrl-µgd*,* Marsrl-Marsd, grrrl-grrd) showed that red light activates the pathways regulating proliferation and growth: cytokinin pathway in microgravity and GRR samples and auxin pathway at Mars *g-*level (GH3 step) but does not alter significantly other hormonal pathways (Appendix A). These results suggest that the gravity level has more impact on hormonal pathways regulation than light conditions.

In summary, an activation of proliferation-promoting pathways (cytokinin and auxin) is evident in microgravity but not at Mars *g-*level. Further activation of the cytokinin pathway was observed with red-light photostimulation (µgrl-µgd comparison). At Mars *g-*level, red light reverses partial inhibition of auxin pathway (GH3 step) which is in agreement with its proliferation-activating effect. On the other hand, stress-related pathways, in particular ABA, ethylene and salicylic acid (SA) seem to be more activated at Mars *g*-level, which could suggest that in partial gravity, the plant perceives the stress signal and responds with activating acclimation mechanisms. 

### 2.4. Plastid and Mitochondrial Genome Expression

We compared the distribution in the genome of DEGs in each condition using ShinyGO analyses [53] and found that in microgravity, the expression of genes encoded in the chloroplastic and mitochondrial genomes were over-represented. This enrichment in plastid and mitochondrial gene expression was specific to microgravity (Figure 4A) and not present in Mars *g-*level. Furthermore, we compared these results with the WT (Ler ecotype) blue-light dataset from the SG experiments (GLDS 251) containing transcriptomic data from seedlings exposed to the following partial gravity levels: microgravity, low gravity level (0.09 ± 0.02 g), Moon gravity (0.18 ± 0.04 g) and Mars gravity (0.36 ± 0.02 g) levels, and found a similar enrichment in chloroplastic and mitochondrial gene expression exclusively in microgravity, but not in any partial gravity including low gravity level (Figure 4B). This enrichment was detected in the upregulated DEGs, specifically 70 chloroplast-encoded genes were upregulated in microgravity dark and 83 in microgravity red-light conditions in our dataset (45 in the blue-light exposed samples from GLDS-251 dataset), and 10 mitochondrial genes were upregulated in microgravity dark and 30 in microgravity red light (5 in the blue dataset). The upregulated chloroplast-encoded transcripts involved multiple subunits of photosystem I and II and NAD(P)H dehydrogenase complex and electron transporters PETA (photosynthetic electron transfer A), PETB and PETD. The upregulated mitochondrion-encoded transcripts involved ribosomal proteins L16 and S3R, cytochrome oxidase 1 and 2 (COX1, COX2), NADH dehydrogenase subunits: 4, 5A and 5C and ATP-binding cassette I2 (ABCI2), a cytochrome C biogenesis protein.

We also compared our results with datasets of two additional spaceflight experiments in which enrichment of organelle-encoded genes was reported in the transcriptomic data available in GeneLab (GLDS-38 and GLDS-44) [29,30]. These experiments were performed using the Biological Research in Canisters (BRIC) hardware without any lighting, so etiolated *A. thaliana* WT Col-0 seedlings grown in microgravity were used. Although not completely, there is a significant overlap in some of these genes in two or more experiment datasets (Appendix A), even considering that it is not uncommon to see differences between spaceflight experiments [54]. The fact that this phenomenon is observed in experiments where different *A. thaliana* WT lines were used and different hardware and environmental conditions were applied, strongly suggests that it is one of the major microgravity effects. These results are consistent with studies of impaired mitochondrial function from drosophila [55] to humans [56].

This plastid and mitochondrial genome expressions were observed in the three different light conditions (blue light, red light, darkness). However, red light leads to generally higher upregulation levels and a greater number of genes were upregulated in our dataset of the red-light photostimulated seedlings compared to plants grown in darkness, or in the Ler dataset of blue-light photostimulated seedlings, as shown by the fold change of the common upregulated genes (Figure 4C). The level of change of all plastid and mitochondrial genes in the three light conditions is shown in Appendix A.

### 2.5. Dissecting the Contribution of the Red-Light Photostimulation to the Response to Each g-Level

To dissect the transcriptomic changes provoked by red-light photostimulation at each *g*-level, we performed an additional set of transcriptomic comparisons of the photostimulated seedlings versus the seedlings grown in darkness at 1*g*, Mars and microgravity levels (grrrl-grrd; Marsrl-Marsd; µgrl-µgd) (Figure 5). The first observation is the fact that the number of upregulated genes was much higher than the number of downregulated at all gravity levels. As expected, we have observed upregulation of genes related to photosynthesis, light harvesting, pigment biosynthesis and response to light stimulus in all gravity conditions including GRR. Other categories upregulated in all gravity conditions included transcripts involved in response to karrikin (which help stimulate seed germination and plant development), phenylpropanoid metabolic processes, secondary metabolic processes and, in microgravity (to a lesser degree in GRR), the reductive pentose phosphate cycle (Calvin cycle). Surprisingly among downregulated transcript categories, we found the ones related to red to far-red signaling pathway and the response to far red light, which indicates that feedback regulatory mechanisms were triggered to tune down the seedling response. Metascape analysis of upregulated transcript heatmap profiles clusters Mars comparison (Marsrl-Marsd) together with GRR comparison (grrrl-grrd)*,* which suggests a more similar plant response in these conditions. In contrast, in downregulated transcript profiles, Mars comparison clusters together with microgravity comparison (µgrrl-µgd). These observations are also reflected in the higher number of upregulated genes common in Mars and GRR than the ones common in Mars and microgravity and a higher number of common genes downregulated between Mars and microgravity than those common for Mars and GRR, or microgravity and GRR (Figure 5). Taken together, these results could suggest that red-light photostimulation has a positive effect on seedlings in Mars gravity level, which is similar as in Earth conditions. Extended heatmaps are shown in Appendix A.

The processes that are the most affected in microgravity by red light include cell wall organization and biogenesis, cell cycle, microtubule-based processes, DNA replication initiation and trichoblast differentiation (Figure 5). This effect is less pronounced in the other gravity conditions, which suggests that red-light photostimulation has especially positive influence on seedlings in microgravity. 

On the other hand, red light has an inhibitory effect on proteolysis in microgravity and Mars conditions, as seen by downregulation of the genes related to amino acid catabolic processes. This observation is in line with the positive effect of red light on biosynthesis and proliferation [14] and is reflected in the increased size of the nucleolus (and meristem) in the red-light photostimulated samples. 

Among the most important transcripts in the cell cycle category upregulated in microgravity conditions were PROLIFERA (*PRL*; Log_2_FC µgrl-µgd 2.16) and multiple cyclins: A2, A3, B1 (B1:1, :3, :4 (cyclin2)), B2(:1, :2, :3), D4 (:1; :2), P4(:1; :2, :3) as well as proliferation markers *AURORA1* and *AURORA2*. Upregulated genes encoding microtubule-related proteins included many subunits of the mitotic spindle, but also some important cytoskeleton-related proteins such as tubulin beta-1 chain (*TUB1*; Log_2_FC µgrl-µgd 1.84) and SPIRAL1-LIKE4 (*SP1L4*; Log_2_FC µgrl-µgd 1.69). *TUB1* encodes beta tubulin regulated by phytochrome A (phyA)-mediated far-red light high-irradiance and the phytochrome B (phyB)-mediated red-light high-irradiance responses [57]. SP1L4 regulates cortical microtubule organization essential for anisotropic cell growth [58].

In terms of cell wall enzymes and proteins, among the upregulated transcripts in microgravity light comparison were present numerous enzymes from the group of hydrolases (*Xyloglucan endotransglucosylase/hydrolase 12*, *XTH12*; *XTH13*; *XTH14*; *XTH20*; *XTH26*; *glycosyl hydrolase 9B13, GH9B13*) and transferases (*galacturonosyltransferase 12, GAUT12*; *rhamnogalacturonan xylosyltransferase 1, RGXT1*; *glucuronoxylan methyltransferase 2, GXM2*) as well as *pectin methylesterase 46* (*PME46*), *pectin acetylesterase 10* (*PAE19*) and *polygalacturonase involved in expansion 1* (*PGX1*). In addition, numerous extensins (*LRX1*, *EXT2, EXT7, EXT8, EXT9, EXT10, EXT12, EXT13, EXT15, EXT16* and *EXT17*) and two fasciclin-like arabinogalactans (*FLA6, FLA7*) were present. These results suggest the cell wall undergoes profound modifications in this condition that might be related to increased cell growth and expansion [54,59].

## 3. Discussion

### 3.1. Red-Light Photostimulation has a Positive Effect on Cell Proliferation in Both Microgravity and Mars Gravity Conditions 

Our morphometric studies indicate that features like meristem and nucleolus size are more robust in the red-light photostimulated seedlings. In the case of the nucleolus, although the difference in size between dark-grown and red-light photostimulated seedlings is not significant in microgravity, the features of nucleolar ultrastructure clearly indicate that red-light photostimulation increases nucleolar activity. Increased root meristem size (expressed as the length and the number of meristematic cells), nucleolar size and changes in nucleolar ultrastructure confirm positive effect of red light at Mars *g-*level, suggesting that the rates of both the meristematic cell proliferation and protein production were increased. On the other hand, the transcriptomic data from different light condition comparisons (grrrl-grrd; Marsrl- Marsd; µgrl-µgd) confirmed that in microgravity, cell cycle and proliferation related genes are upregulated and the hormonal routes promoting proliferation activated (Figure 5B and Appendix A). 

Red and far-red light are perceived by photoreceptors termed phytochromes, which are expressed in different zones of the root (reviewed in [60]). PhyA and phyB, expressed mainly in the root tip, are involved in both red-light-induced positive root phototropism and gravitropism [61,62]. Red-light photostimulation of seedlings in GRR has a positive effect on the overall physiology in comparison to the seedlings kept in darkness for the last two days of cultivation [37]. In addition, red light is known to stimulate cell proliferation and ribosome biogenesis [14], which are observed in our results, particularly when the gravity vector cannot completely guide the plant development. In addition, red light also restored the meristematic competence balance, which was extensively described to be affected in early plant development in our previous ROOT experiment in the ISS in etiolated *A. thaliana* seedlings [3].

Thus, we conclude that red-light photostimulation could help plants to overcome some of the deleterious effects of the spaceflight environment. Similar effects were seen previously in our experiments with blue-light photostimulation [22].

### 3.2. Microgravity has a Deleterious Effect on Plant Physiology: Elevated Plastid and Mitochondrial Genome Expression is Observed in Microgravity, but Not in Partial Gravity

Dysregulation of the genes involved in photosynthesis was specific for microgravity condition. Photosynthesis-related genes were upregulated in both light conditions, which was related to the increased plastid genome expression (to be discussed further). On the other hand, in the µgrl sample, genes involved in photosynthesis were also downregulated. This is in agreement with the results of the previous SG experiment [21], although only 16 genes were common for both datasets obtained from different light conditions. Reduction in photosynthesis activity, and specifically in photosystem I complex, was observed in previous studies of *Brassica rapa* plants grown in space [63], and in *Oryza sativa* plants grown in simulated microgravity [64]. Furthermore, structural changes in chloroplasts, such as alterations of thylakoid membranes in seedlings grown in real [65] and simulated microgravity [66] were reported, as well as a reduction of chloroplast size in simulated microgravity [67]. Downregulation of LHDB proteins, which was observed in microgravity in red- (our dataset) and blue-light stimulated seedlings (GLDS-251) (Appendix A), is known to affect stomatal closure in effect reducing photosynthetic activity. This downregulation also decreases plant tolerance to drought stress [68,69]. 

Dark-grown samples showed a general decrease in photosynthetic activity as demonstrated by the transcriptomic analysis of 1*g* GRR samples [37]. It is therefore not surprising that we have not observed this category in dysregulated genes in µgd-grrd comparison. 

Additionally, few ERFs important for development of tolerance to a number of abiotic stressors were downregulated in microgravity: *Translucent Green* (*TG*), downregulated in microgravity and darkness, is involved in drought tolerance [70], *C-repeat/DRE binding factor 1* (*CBF1*) and *CBF2*, downregulated in microgravity in both light conditions, are involved in tolerance to freezing [71,72], *AtERF72*, also downregulated in both light conditions, is involved in the tolerance to peroxide (H_2_O_2_) and heat stress [73]. In summary, the downregulation of these ERFs in microgravity may have a negative effect on *A. thaliana* tolerance to adverse conditions. 

Two organelles seem to be particularly affected in microgravity during space flight. Plastids contain 3000–4000 proteins and most of them are encoded in the nucleus [74]. Around 90 to 100 are encoded in the chloroplast genome [75]. The majority of these genes are upregulated in microgravity (70 in µgd and 83 µgrl). Mitochondria contain around 3000 proteins, from which 57 were identified in the mitochondrial genome [10,76]. In our dataset, we identified 10 of them in µgd, peaking to 30 out of 57 in µgrl and 10 in microgravity and darkness. Various factors influence plastid gene expression, such as light, temperature, plastid development or circadian clock [74]. Abscisic acid (ABA) represses transcription of chloroplast genes [77]. The ABA pathway was activated in the Mars samples, which could explain why this phenomenon is not observed in this condition. Upregulation of plastid-encoded genes was reported before in spaceflight experiments [29].

Mitochondria and chloroplasts are tightly involved in cellular metabolism and are thought to be initial sensors for cellular dysfunction caused by external stress. Research to date suggests that factors that participate in signaling between these organelles and the nucleus (anterograde communication: communication from nucleus to organelle; retrograde communication: from organelle to nucleus) also participate in the recognition of the stress level. The decision is whether to adjust the metabolism, or to execute programmed cell death (PCD) [78]. The key signaling molecules in mitochondrial dysfunction are ANAC017, ANAC013 and Alternative Oxidase 1a (AOX1a) [79]. *ANAC017* is only slightly downregulated in microgravity and *AOX1a* and *ANAC13* is only upregulated in Marsrl condition (around 1-fold). In addition, there is a set of mitochondrial proteins which are consistently upregulated in stress conditions when the dysfunction of mitochondria takes place [80,81,82], but only one of these is upregulated in the µgrl sample (*AT3G50930*) and three downregulated (µgd, µgrl: *AT1G20350*, *AT1G21400*; µgd: *AT4G15690*). 

Based on our results, it would seem that there is a dysregulation of the chloroplast and mitochondrial genome expression, and this dysregulation is not perceived and corrected in the typical retrograde communication in response to organelle dysfunction. Moreover, this organelle dysfunction is not present in partial gravity level, probably because the retrograde communication is not disturbed. Supporting the last assumption is the fact that Sigma factor binding protein (SIB1), which binds sig1R factor (nuclear encoded factor that regulates the chloroplast genome expression) and has an important role in the retrograde communication, is downregulated in µgd, but upregulated in Mars conditions. *At*WRKY40 was shown to be a repressor of retrograde-mediated expression while *At*WRKY63 has an opposite activating effect [79]. The antagonistic functioning of WRKY40 and WRKY63 could also play a role in this dysregulation. At Mars *g*-level, *AtWRKY40* is upregulated, subduing the expression of stress-responsive genes (genes responding to mitochondrial and chloroplast dysfunction). On the other hand, upregulation of *AtWRKY63* could play a role in the dysfunction of both organelles. It has been reported that disturbed mitochondrial retrograde signaling leads to increased sensitivity of plants to stress conditions [83]. Mitochondrial retrograde signaling is involved in acclimation to flooding, and *At*WRKY40 is involved in promoting this acclimation together with *At*WRKY45 [83], which is also upregulated at Mars *g*-level and downregulated in microgravity. 

Mitochondria and WRKY40 also participate in response to touch and wounding. This response involves other signaling factors such as OM66 and mitochondrial dicarboxylate carriers DIC2, DIC1 [84,85]. The *OM66*, *AtWRKY40*, *DIC1* and *DIC2* are upregulated at the Mars *g*-level (both light conditions). It is possible that at the Mars *g-*level, the seedling is activating the response to touch in the search for the direction stimuli, whereas in microgravity this route is not activated (as *OM66*, *AtWRKY40, DIC1* and *DIC2* are not upregulated).

However, the mechanism involved in the perception of microgravity and the mechanism that causes the dysregulation of the organellar genome transcription remains unresolved. Nevertheless, our results show that by the combination of light with applying even low *g-*level, this stress response observed at the Mars *g*-level (and even more intense at the Moon *g*-level [18,22]) can be corrected. 

### 3.3. Seedlings Grown at Mars g-Level Activate Stress Responses Involving WRKY TFs Possibly Leading to Acclimation

The acceleration similar to the Mars *g*-level in the EMCS centrifuge in orbit is enough to provide gravitropism cue to plants on the ISS as seen in Figure 6 and previously reported [15]. Consistently, the transcriptome changes observed at Mars *g-*level are very different from those found in *A. thaliana* exposed to microgravity, which is especially clear in the seedlings grown without photostimulation. DEGs involved in multiple stress responses, like hypoxia (decreased oxygen levels), drought, reactive oxygen species, osmotic and biotic stresses are altered in both gravity levels. These observations are in accordance with previous experiments in the microgravity and partial gravity conditions [22,25,54]. However, when we compare the number of common DEGs in microgravity and Mars conditions (Figure 2), we observe only 15 upregulated genes and 55 downregulated genes. In fact, even though common GO categories are upregulated and downregulated for both conditions, only a few or none of the specific DEGs are common for both gravity levels (Appendix A). Moreover, photosynthesis, which is highly affected by microgravity [21], does not seem to be disturbed at Mars *g-*level. It is evident that both conditions induce different responses in seedlings and, therefore, the strategies to grow plants, either during spaceflight or on a planet with reduced gravity, but enough in magnitude to trigger a full gravitropic response, should also be different.

GO categories common for all the samples are most likely related to the spaceflight conditions; however, the plant responds to these environmental factors differently in microgravity and Mars *g*-level, as suggested by the low number of common DEGs. The bioavailability of oxygen in the spaceflight environment is reduced and very dependent on the hardware used for the experiment in orbit, provoking the plant response to hypoxia [54,86]. This stress is closely related to waterlogging response (or water stress), also present in the upregulated group in the GO analysis; in fact, when a plant is submerged under water the availability of oxygen is reduced and response to hypoxia activated [87]. Morphological changes typical for plants in response to flooding, such as the appearance of adventitious roots in *A. thaliana* (“roots on the stem”; [87]), were observed in BRIC-16-Cyt experiment [88]. In addition, Stout et al. [89] demonstrated increased activity of fermentative enzymes in the roots of *B. rapa* grown in the space environment, which indicates root zone hypoxia. It is possible that the plant activates these known mechanisms to enhance oxygen intake. The response to the osmotic stress is also a category frequently dysregulated in space experiments [29]. The nature of this response is not well understood but it was suggested that plants could activate in microgravity the response to osmotic stress due to the absence of structural guide, compensating it with the stabilization of microtubules [29,90]. Nevertheless, the response to this stressor is also present in partial gravity where the seedling seems to perceive gravitational cues (Figure 6). Our results suggest that the hormonal and transcriptional routes involved in response to osmotic stress, such as the ABA pathway and ERF TFs upregulation, are activated. ERFs are characterized by the presence of ERF DNA binding domain [91] and fulfill a wide range of functions in response to multiple stresses. Differential expression of ERFs was reported in adverse conditions such as waterlogging and hypoxia [92,93]. 

At the Mars *g*-level, we observed a clear enrichment in WRKY TFs. They are involved in a wide variety of functions from abiotic and biotic stress response to developmental and multiple physiological processes on Earth [83,94,95,96,97]; reviewed in [47]. They also participate in hormonal response for example in JA/SA hormonal signaling. The WRKY family is defined by the presence of at least one WRKY DNA binding domain (DBD). They interact with W-box (with TTGACC/T motif) and clustered W-boxes located in the target genes that are activated or repressed under a specific condition. Most WRKY TFs are multifunctional meaning they play a role in a number of responses (see Table 1) thanks to multiple functional domains they contain (zinc-finger motifs, leucine zippers, kinase domain, (CaM)-binding domain etc.) [47,98,99]. For example, *AtWRKY75* which is upregulated at the Mars gravity level, plays a role in the immune process, response to osmotic stress, regulation of phosphorus deficiency signaling, development of the root hairs and has a positive effect on leaf senescence [100]. 

The multiple functional domains that each WRKY contains enable them to form complexes with numerous proteins and fulfil a wide range of functions [47]. This multifunctionality makes WRKY TFs a perfect target for genetic manipulation to create more resistant breeds that can be used in future space experiments. By modifying just one WRKY, a resistance to a set of abiotic and biotic stresses and developmental traits can be achieved. Since TFs such as WRKY are key players in molecular breeding of crops due to their important role in the process of crop domestication [101,102], they have received a lot of attention in recent years. Genetic modification to obtain a specific positive trait was successfully used before, for example, to produce drought-resistant rice overexpressing *Os*WRKY30/70 [103,104]. Similar strategies could be applied also to develop cultivars for “space farming.”

The upregulation of WRKY and other important TFs families, such as NACs, in Mars samples suggests that seedlings grown in partial gravity level activate multiple routes to cope with stress associated with space environment, and they can acclimate by modulating genome expression. On the other hand, among the genes upregulated in microgravity, a lower number of TFs can be found. In fact, as mentioned before, a number of ERFs which participate in development of tolerance to various adverse factors are downregulated in this condition.

Hormonal pathways promoting growth and proliferation are activated in microgravity and hormonal pathways promoting stress response are activated at Mars *g*-level

A summary of anatomical changes and changes in hormonal pathways is presented in Table 2 and Table 3. Auxin and cytokinin pathways, two hormones regulating cell growth and proliferation, were disrupted differently in microgravity and Mars *g*-level. Our results suggest that both pathways were activated in the microgravity conditions, and few elements of these routes were repressed at Mars *g-*level. Dysregulation of auxin pathway in space-grown seedlings was reported in previous studies [23,29].

The SA pathway was activated in red light at Mars *g* and inhibited in the microgravity darkness conditions, and the JA pathway was inhibited through JAZ proteins in all samples. Both SA and JA play major roles in the defense response to pathogens. The role of SA is to activate the resistance against biotrophic pathogens, whereas JA is involved in the activation of defense mechanisms [105]. Specifically, the principal function of JA is the promotion of the resistance to plant pathogens by production of defense compounds and, at the same time, it inhibits plant growth. Both, JA and SA, also participate in the response to abiotic stressors and development of tolerance. JA is involved in response to cold, drought, salinity and light (reviewed in [106]). SA was reported to be involved in response to environmental stressors such as high and low temperature, drought, salinity and UV-B radiation (reviewed in [107]). SA-mediated mechanisms together with reactive oxygen species (ROS) and glutathione (GSH) regulate the transcription of different sets of defense genes in a spatio-temporal manner [108]. On the other hand, JA regulatory pathway act through the crosstalk with other phytohormone pathways including ABA, SA and ethylene [106] and is fine-tuned by numerous JA compounds and their different modes of action [109]. Hormones that promote plant growth, such as auxin, gibberellins and cytokinins, repress JA/SA mediated defense-response to prioritize the growth. On the other hand, activation of SA and JA routes can suppress these growth-promoting pathways to activate the defense [110]. 

JAZ proteins, which were upregulated in both gravity conditions, are transcriptional repressors in JA signaling [111], and by tuning down the response to JA, they enable the recovery of the organ growth [112]. They act on diverse TF families including bHLH, MYB or WRKY [113]. Furthermore, they interact with DELLA proteins to regulate JA and Gibberellin acid (GA) signaling [114], which leads to regulation of plant growth and plant defense response upon environmental conditions. JAZ1 was upregulated at Mars gravity and JAZ4 in microgravity. Apart from the leading role in plant resistance and defense, JAZ proteins are also implicated in the response to abiotic stresses. JAZ1 confers tolerance to alkaline stress [115] and JAZ4, which is not induced as other JAZ proteins by insects or wounding [116], was shown to be involved in control of leaf senescence [117], freezing tolerance [118], growth and development [113]. AIB, which interconnects JA-ABA pathways, was downregulated in microgravity conditions. AIB interacts with JAZ proteins to negatively regulate jasmonate responses. It is induced by ABA and participates in developing a drought tolerance [49]. The ABA pathway is strongly activated at Mars *g-*level. It is the most important regulator of the response to drought and osmotic stress in plants and a positive regulator of root hydrotropism [119]. ABA, together with MAP-kinase (MAPK) perception and signaling pathways, are involved in all abiotic stresses which cause the decrease of turgor pressure and water loss [48].

The ethylene pathway, together with ERFs, are involved in a wide range of stress responses. Although it has been suggested that the detection of an ethylene response in spaceflight experiments was an effect of the ethylene accumulated from the previous experiments, any ethylene accumulation during this experiment would have been erased from the hardware by a flushing procedure performed before the initiation of the experiment as well as by constant air flow by a connected gas removal module [20,21,120]. 

In summary, the dysregulation of different hormonal routes in our samples suggests that, in microgravity, the seedlings do not address the external stress and take the option of growing by activating growth and proliferation promoting auxin and cytokinin routes. On the other hand, in the Mars *g* samples, the growth-promoting routes are inhibited and ABA, ethylene and salicylic acid routes, known for their crucial role in response to osmotic, drought and biotic stresses, are activated. 

## 4. Materials and Methods 

### 4.1. Spaceflight Experiment and Procedures

The SG experiments were a series of spaceflight experiments aimed at investigating the response of young seedlings of *A. thaliana* to the joint stimuli of different levels of gravity and light. Here, we present results corresponding to part 2 and part 3 of the SG series (SG2 and SG3 experiments).

Experimental containers (ECs), each containing 5 culture chambers (cassettes) with 28 seeds attached to gridded nitrocellulose membrane with guar gum (as described in [37,121]) were used. SG2 was sent to the International Space Station (ISS) during the SpaceX CRS-4 (September 2014) and returned on CRS-5 (February 2015), and SG3 was sent to the ISS during the SpaceX CRS-11 campaign (June 2017) and returned a month later, on the same mission.

The experiment did not start until the ECs were loaded into the EMCS and the cassettes were hydrated [20]. Experimental conditions such as hydration, environmental humidity (>80%), gas exchange (O_2_ levels kept at 10%; CO_2_ at 0.45%) and temperature (22.5 °C *±* 2 °C), were monitored and controlled remotely from the Norwegian User Support and Operation Centre (N-USOC, Trondheim, Norway). The GRR was performed at this site in the identical hardware using the same experimental conditions a few months later. During the experiment, seedlings were grown for four days in a long day photoperiod (16 h white light, 30–40 μmol/m^2^s and 8 h darkness) at two nominal *g*-levels on board the ISS (microgravity and Mars gravity level nominally 0.3 *g* (0.34 ± 0.05 *g*), as provided by the EMCS centrifuge) and the Ground Reference Run 1*g* control. In the last two days of the timeline of the experiment, a change in the light conditions was introduced, half of the material was photostimulated with red light-emitting diodes (LEDs) on one lateral of the ECs, and the rest of the material was grown in darkness.

The timeline of the experiment is shown in Figure 6. The three gravity levels were constant for each EC throughout the duration of the experiment. Before the experiment began, flushing of the EMCS was performed to erase any possible traces of ethylene and other gases from previous experiments. When the six days of seedling growth were completed, the ISS astronauts then removed each EC from the EMCS and froze the samples at −80 °C in orbit (in the MELFI) or used the FixBox to fix the samples with aldehydes for morphological studies (as described in [37]). Germination rate was calculated as the percentage of germinated seeds.

### 4.2. Confocal Microscopy

The details of the spaceflight device (termed the FixBox) used for fixation of the samples in 5% (*w*/*v*) formaldehyde and the procedure are described in [122]. Briefly, seedlings were fixed in 5% (*w*/*v*) FA for 3 h at room temperature (RT) and then kept at 4 °C until return to Earth (µg and Mars *g*-level) or directly processed in the GRR. Fixative was rinsed three times in PBS and then seedlings were digested with digestion solution, containing: 2% (*w*/*v*) cellulase, 1% (*w*/*v*) pectinase, 0.05% (*w*/*v*) macerozyme, 0.4% (*w*/*v*) mannitol, 10% (*v*/*v*) glycerol and 0.2% (*v*/*v*) Triton x-100 in PBS for immunofluorescence with an anti-fibrillarin antibody (Abcam, Cambridge, UK, ab4566 [38F3]). For cell wall staining, the tissues were digested with enzymes not containing cellulose and with 0.5% macerozyme (*w*/*v*) and stained with SCRI 2200 a cellulose specific stain (Renaissance Chemicals, North Duffield, UK) [123,124]. Fibrillarin area and meristem size were measured with ImageJ v1.53c. Statistical analyses of the measurements were made using SPSS v25 software.

### 4.3. Electron Microscopy

The FixBox [122] was also used for fixation of the samples with 4.5% (*v*/*v*) glutaraldehyde (Sigma-Aldrich, St. Louis, MO, USA, #G5882) and 1.5% (*w*/*v*) formaldehyde (Electron Microscopy Sciences, Hatfield, PA, USA, #15710) in PBS for electron microscopy analysis. Following aldehyde fixation, samples were post-fixed in 1% (*w*/*v*) osmium tetroxide in PBS for 1 h and dehydrated in ethanol. Root tips were embedded in epoxy resin and then sectioned. Ultrathin sections were mounted on nickel grids coated with a 0.5% (*w*/*v*) Formvar film (Sigma-Aldrich, St. Louis, MO, USA, #09823) and stained with 5% (*w*/*v*) uranyl acetate (Electron Microscopy Sciences, Hatfield, PA, USA, #22400) and 0.3% (*w*/*v*) lead citrate (Electron Microscopy Sciences, Hatfield, PA, USA, #17800). Next, the samples were examined in a JEOL 1230 transmission electron microscope (TEM) at 80 kV. 

### 4.4. RNA Extraction and Sequencing

Details of RNA extraction and sequencing are described in [37]. Briefly, the RNA extraction kit MACHEREY-NAGEL (Macherey-Nagel (MN), Düren, Germany, #740949) was used to extract RNA for pools of 8–10 seedlings. The RNA extraction kit includes DNAse treatment for 15 min. RNA quality was analyzed with the Bioanalyzer 2100 expert Plant RNA nano with Agilent RNA 6000 Nano Kit (Agilent Technologies, Santa Clara, CA, USA, #5067-1511). Sequencing was performed on the Illumina HiSeq2500 sequencer (Center for Genomic Regulation, Barcelona, Spain) with stranded RNA read type and 50 bp read length. Seventeen total RNA samples were used to generate sequencing libraries using the Illumina TruSeq RNA Library Preparation Kit (Illumina, San Diego, CA, USA, #RS-122-2001). Samples were individually indexed. The samples then were combined at equimolar proportions into two pools. Each pool was loaded onto two lanes of a flow cell. Sequencing was performed until the 25 million reads per sample objective were reached (27.5 ± 1 millions of sequences obtained). Results from these studies have been deposited as the GLDS-314 and are available at NASA’s GENELAB repository (DOI:10.26030/z5yf-jx91, https://genelab-data.ndc.nasa.gov/genelab/accession/GLDS-314, [40]).

### 4.5. Functional Analysis

Differential expression analysis was done using Deseq2 [125]. PCA was made using iDEP.91 [126]. Gene Ontology analysis of the DEGs was done using ShinyGO [53] with default settings and Metascape [127] with custom analysis adding molecular function and cellular component. ShinyGO was also used to analyze the distribution of query genes across the genome. Venn diagrams were made using jvenn [128]. String v11.0 was used for protein–protein interaction analysis [129]. Functional analysis of the dysregulated genes involved in phytohormone signaling pathway was performed using KEGG Pathway, the reference database for pathway mapping in KEGG MAPPER [130] available at Kyoto Encyclopedia of Genes and Genomes (KEGG). For the analysis of enrichment in a comparison of a specific gene family, a Chi-squared test was used using GraphPad software v5 (San Diego, CA, USA). The list of the identifiers and Log_2_FC values of the genes dysregulated in microgravity (Appendix A) and Mars *g*-level (Appendix A) are given in the Appendix A.

## 5. Conclusions

We conclude that the response to reduced gravity does not show a gradual decrease in the intensity of the effects observed at microgravity, but clearly differentiated effects on plant growth and physiology are detected, as shown by anatomical and transcriptomic changes. In microgravity, *A. thaliana* accelerates cell proliferation and growth in the root meristem, even though some of the cellular processes, such as retrograde and anterograde communication, appear to be disturbed. This strategy could be activated by applying alternative directional cues, such as light (in particular, red light), and it could lead to adverse effects on the long-term plant development, considering the high energetic cost that it entails. On the other hand, at the Mars gravity level, the seedling perceives external stress and activates responses cooperating with the acclimation of the plant to the environmental conditions, such as upregulation of WRKY TFs. Red light increases cell proliferation at all gravity levels, as shown by microscopic and transcriptional analyses and it is particularly required to prime the adaptive stress response to the Mars *g*-level. In long-term applications, the combination of partial gravity level and red-light photostimulation could be used in space farming to avoid dysregulation of those pathways appearing affected in microgravity and to promote robust seedling growth.

## Figures and Tables

**Figure 1 ijms-22-00899-f001:**
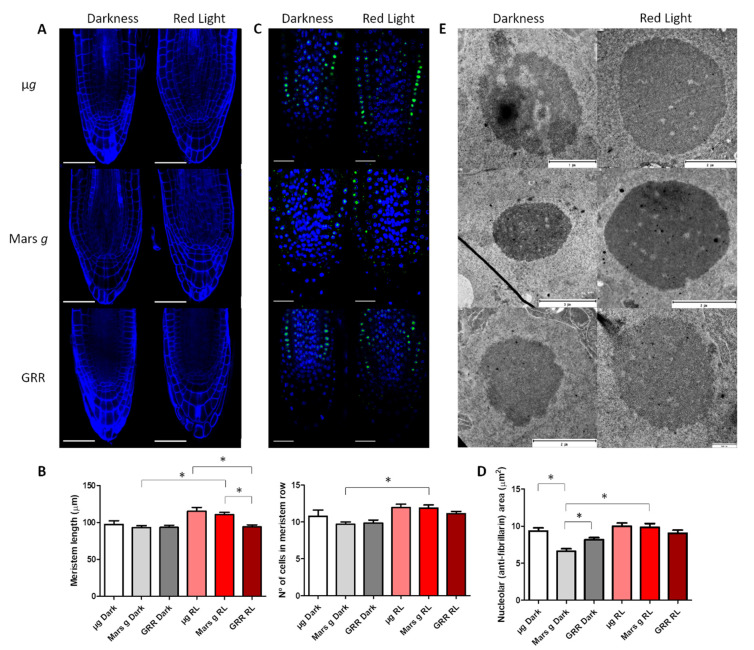
Meristem and nucleolus effects of microgravity and Mars gravity. (**A**) Confocal microscope images of cell-wall-stained root meristems of the different gravity and photostimulation conditions. Scale bar represents 50 µm. (**B**) Quantification of meristem length and number of cells per meristematic layer. Meristem length statistical analysis was made with ANOVA with Scheffe test post-hoc. Number of cells in the meristem row analysis was ANOVA with T3-Dunnett post-hoc (data not homoscedastic). In both cases the *p*-value for significance is 0.05. (**C**) Confocal images of immunostained root meristems; green: anti-fibrillarin, blue: DAPI. Scale bar represents 25 µm. (**D**) Nucleolar area (anti-fibrillarin immunostaining quantification) in µm^2^. Statistical analysis was made using a non-parametric test with corrected *p*-value of 0.0033 (data without normal distribution. Bars represent mean + SED. * indicates significant statistical differences. (**E**) Electron microscope images of meristematic cell nucleoli. The difference in nucleolar size among different conditions is clearly observed. Furthermore, typical structural models corresponding to inactive nucleoli are seen in darkness conditions, especially in microgravity.

**Figure 2 ijms-22-00899-f002:**
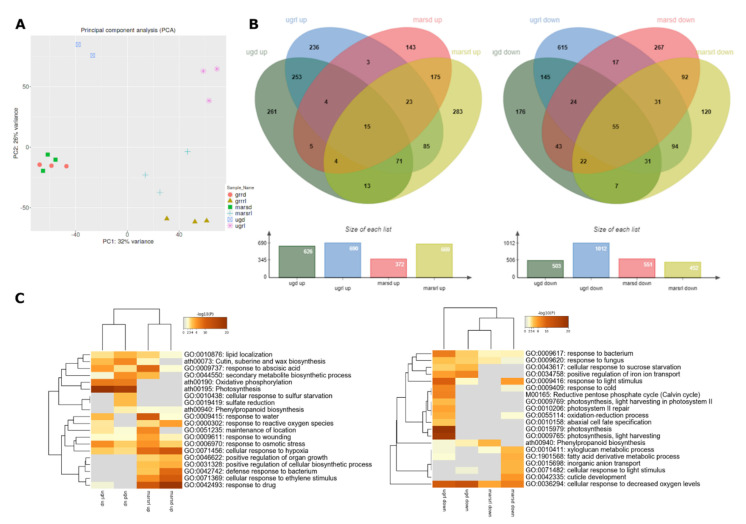
Global transcriptomic effects of microgravity and Mars gravity. (**A**) Principal component analysis (PCA) where all replicates included in RNASeq analysis are represented. (**B**) Venn diagrams with differentially expressed genes (DEGs) with q-value < 0.05 and Log_2_FC > 1.5 (and < −1.5), separated in upregulated (**left**) and downregulated (**right**) genes. (**C**) Metascape Gene Ontology Heatmaps of top 20 enriched clusters for upregulated (**left**) and downregulated (**right**) DEGs.

**Figure 3 ijms-22-00899-f003:**
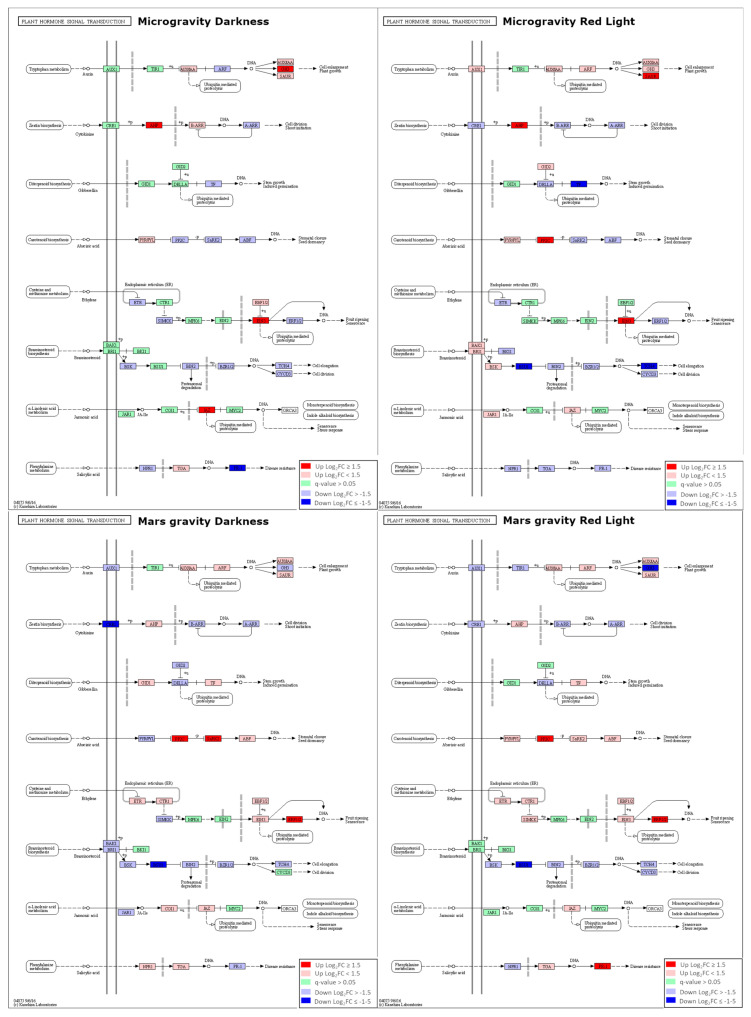
Hormone signaling changes in microgravity and Mars gravity. Plant hormone signal transduction (ath04075) Kyoto Encyclopedia of Genes and Genomes (KEGG) representation with color-coded changes in each experimental condition: microgravity darkness (µgd-grrd), microgravity red light (µgrl-grrrl), Mars gravity darkness (marsd-grrd) and Mars gravity red light (marsrl-grrrl).

**Figure 4 ijms-22-00899-f004:**
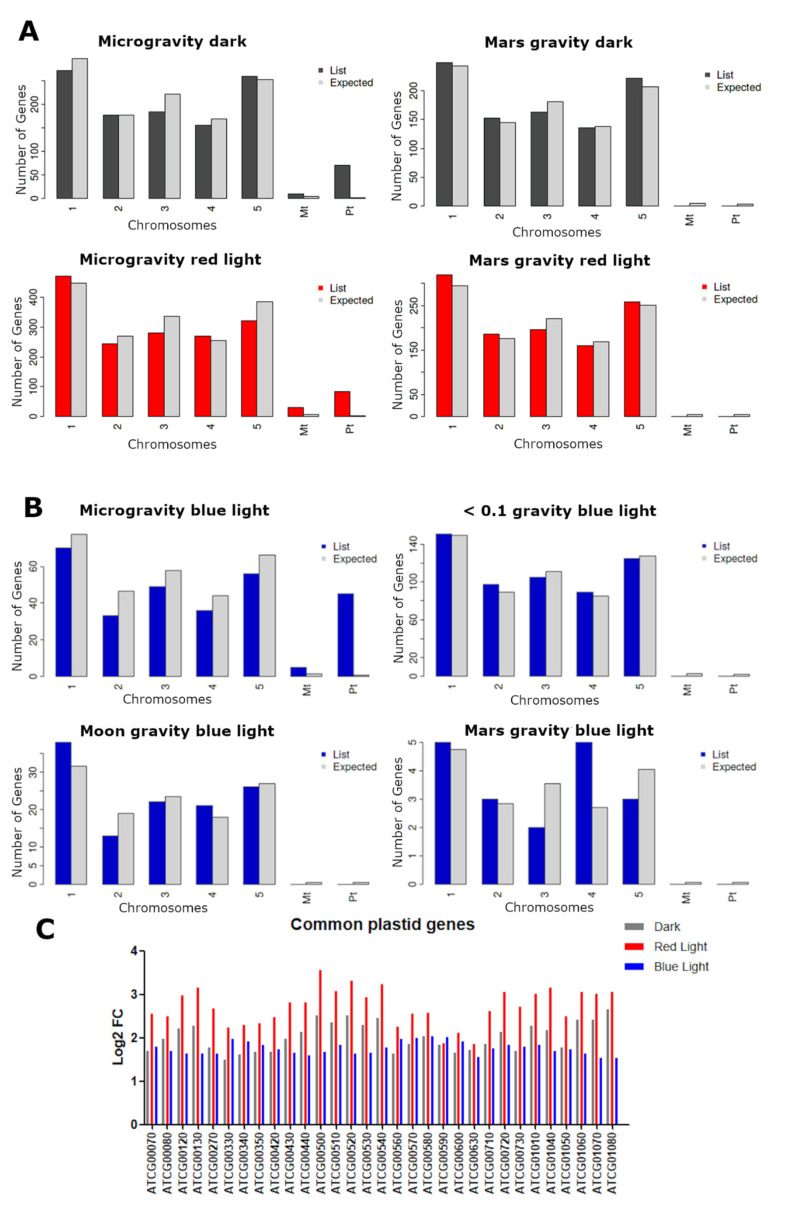
Plastid and mitochondrial genome expression in microgravity. (**A**) Distribution of DEGs across chromosomes in different comparisons: µgd-grrd, marsd-grrd, µgrl-grrrl and marsrl-grrrl. (**B**) Distribution of DEGs in GLDS-251 (NASA GeneLab Accession No.) blue-light stimulated seedlings in different gravity levels: microgravity, <0.1*g*, Moon gravity and Mars gravity. Bars (**A**,**B**) represent number of genes in query list (dark grey, red or blue) and expected number of genes (light grey). The distribution of the query genes is statistically significant (Chi-squared test) in microgravity dark (*p*-value: 1.1 ··· 10^−209^), microgravity red light (*p*-value: 1.5 ··· 10^−207^), mars red light (*p*-value: 0.014), microgravity blue light (*p*-value: 0). (**C**) Log_2_FC of the common genes of the three microgravity comparisons: µgd-grrd (grey), µgrl-grrrl (red) and GLDS251 µ*g*-1*g* control with blue light stimulation (blue).

**Figure 5 ijms-22-00899-f005:**
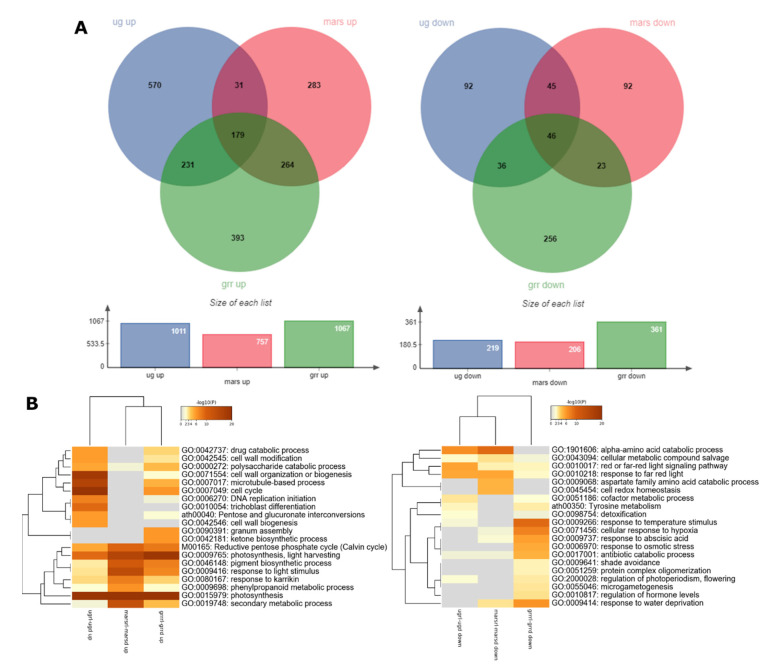
Red-Light induced changes in gene expression across gravity conditions. (**A**) Venn diagrams representing DEGs (with q-value < 0.05 and Log_2_FC > 1.5 (or <−1.5). Left: upregulated genes. Right: downregulated genes. (**B**) Metascape Gene Ontology (GO) heatmaps of top 20 enriched clusters. Left: upregulated genes. Right: downregulated genes.

**Figure 6 ijms-22-00899-f006:**
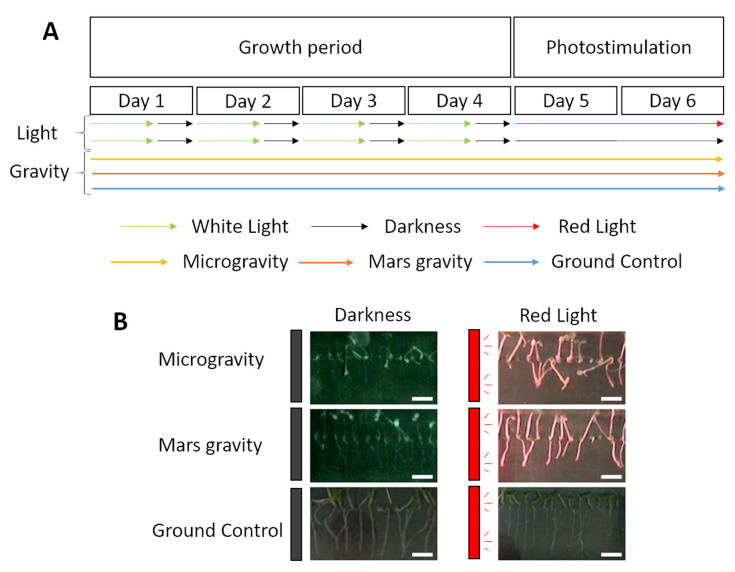
Experiment design. (**A**) Experimental timeline. Seedlings grown during six days in a long day photoperiod regime followed by two days of red-light photostimulation or darkness. (**B**) Images of seedlings at the end of the experiment. Grey and red rectangles at the left of the images represent light-emitting diode (LED) light position for photostimulation. Scale bar represents 3 mm.

**Table 1 ijms-22-00899-t001:** Upregulated WRKY-domain transcription factors (TFs) in Mars gravity level. Statistically significant (q-value < 0.05) WRKY TF in marsd-grrd and marsrl-grrrl comparisons. Log_2_FC for each TF in each comparison is shown. Reference list from this table is provided in the Appendix A.

Name	Group	Functions	Marsd-grrd	Marsrl-grrrl
*At*WRKY38	III	- negative roles in plant defense [1] - involved in SA signaling pathway [1]	2.34	2.19
*At*WRKY40	II-a	- JA-signaling repressor [2] - role in response to salinity/osmotic [3] - response to touch [4] - defense response	2.13	1.82
*At*WRKY45	I	- flooding stress [5] - phosphate ion transport [6]	1.19	0.81
*At*WRKY46	III	- regulates development, stress and hormonal response by facilitating growth of lateral roots in osmotic/salt stress through ABA signaling and auxin homeostasis [7] - role in the immune process; induced by *P. syringae* or SA [8] - cellular response to hypoxia [9]	3.73	3.61
*At*WRKY51	III	- mediates SA- and low oleic acid-dependent repression of JA signaling involved in plant defense [10]	1.85	2.14
*At*WRKY53	III	- positive effect on plant senescence [11] - role in the immune process - may play a role in SA signaling pathway [8]	2.35	1.58
*At*WRKY54	III	- negative regulation of senescence [12] - defense response - regulation of brassinosteroid, JA, SA and ethylene pathways (arabidopsis.org) - osmotic stress	2.82	2.00
*At*WRKY59	II-c	- regulation of transcription	4.34	3.02
*At*WRKY62	III	- induced by *P. syringae* or SA, [1] - negative role in plant defense.	2.77	2.48
*At*WRKY66	III	- regulation of transcription	1.95	3.56
*At*WRKY75	II-c	- (with AtWRKY44) development of the root hairs [13] - has a positive effect on leaf senescence [14] - participate in the regulation of phosphorus deficiency signaling [13]	1.78	1.61
*At*WRKY33	I_C	- binds to SIB1, JAZ1 and JAZ5 affecting JA-mediated defense signal pathway [15] - involved in abiotic stress response, in particular salt/osmotic stress (arabidopsis.org)	1.84	1.45
*At*WRKY70	III	- negative regulation of senescence [12] - role in the immune process may play a role in SA signaling pathway [8]	2.28	1.49

**Table 2 ijms-22-00899-t002:** Summary of anatomic changes in the root compared to the corresponding ground control. Arrow pointing up: increase. Arrow pointing down: decrease. Statistically significant changes are highlighted in red.

		Meristem Length	Meristem No. of Cells	Fibrillarin Area	Nucleolar Ultrastructure
Microgravity	Darkness	↑	↑	↑	↓
	Red Light	↑	↑	↑	↑
Mars gravity	Darkness	-	-	↓	↑
	Red Light	↑	↑	↑	↑

**Table 3 ijms-22-00899-t003:** Summary of phytohormone signaling changes in gene expression of each condition compared to the ground control. Arrow pointing up: upregulation. Arrow pointing down: downregulation. Red indicates stronger changes according to fold change (at least one step Log_2_FC > 1.5).

		Auxin	CK	Brassinosteroids	ABA	Ethylene	JA	SA
Microgravity	Darkness	↑	↑	↓	↓	↑	↑	↓
	Red Light	↑	↑	↓	↑	↑	↑	↓
Mars gravity	Darkness	↓	↓	↓	↑	↑	↑	↑
	Red Light	↓	↓	↓	↑	↑	-	↑

## Data Availability

The original sequencing data described in this study have been deposited at NASA’s GENELAB repository as the GLDS-314 dataset (DOI: 10.26030/z5yf-jx91, https://genelab-data.ndc.nasa.gov/genelab/accession/GLDS-314, [40]).

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
