# Peer review of "From Spaceflight to Mars g-Levels: Adaptive Response of A. Thaliana Seedlings in a Reduced Gravity Environment Is Enhanced by Red-Light Photostimulation"

_ijms, 2021, doi:10.3390/ijms22020899_

Round 1
Reviewer 1 Report
The manuscript entitles “From spaceflight to Mars g-levels: adaptive response of A. thaliana seedlings in a reduced gravity environment is enhanced by red light photostimulation”, proposed for publication in “International Journal of Molecular Sciences”, comprises an interesting work regarding the development of A. thaliana plants in different gravity conditions, focusing on root growth and transcriptomic responses. Although it is work developed with the adequate methodologies, some aspects should be included, mainly about the statistical analysis and the reason of the use of too young seedlings (6 days) instead of plants growth for a longer period. In addition, most of the figures have a poor quality and are not legible, and consequently the described results can not be confirmed in them. Consequently, I consider that this work should be revised again with major modifications before considering it for publication.
In the following lines, more minor details are included to improve manuscript quality.
Introduction
L. 48 and 99: Change “exposed” to “subjected”.
Results
L. 105-111: This part and the Figure 1A should be moved to “Materials and Methods”.
Fig. 2: The quality of the images should be improved specially in the Figure 2B and 2D, where it is difficult to observe which groups are statistically different. Indicate in the caption the statistical analysis used for considering statistical differences. Why was not an ANOVA analysis used?
The quality of figures 3, 4, 5 and 6 should be also improved, since they are not readable.
Figures 5A and 5B should include axis names.
Table 1. The names of the transcription factors, should include ”At” in italics. Revise it also in the text.
Revise the format of accessions. For example, in line 301 appears “At5g65100” and in the line 312 “AT1G50060”.
Discussion
Table 3: Change “pytohormone” to “phytohormone” in the caption. How was the indicated statistical analysis performed in gene expression analyses?
L622-628: SA and JA, as ABA, are also involved in abiotic stress signalling. You should revise in the bibliography and discuss deeply the importance of these hormones in response to tropisms.
Materials and Methods
Why was the experiment performed with Young plants of 6-days old? It would be interesting to see how the growth of these plants is affected by these gravity conditions in longer periods.
L. 665-675: Is there any additional information about the values of gas exchange, as O2 or CO2 concentrations?
L. 702-713: Was the extracted RNA treated with DNAse to remove the possible extracted DNA and ensure RNA quality?
Conclusions
L. 739: Change “can” to “could” since this work, with 6-day old plants can not be considered a long-term application as indicated in this sentence.
Author Response
Response to the Reviewer 1
First of all, we would like to thank the reviewer for the positive and constructive critiques. After following the recommendations of the reviewer, we are convinced that the paper has improved, and the results reported will reach the journal’s readership more efficiently.
We have classified the criticisms, suggestions and comments expressed by Reviewer 1 into four categories that will be answered separately:
- Quality of Figures
- Use of seedlings (six-day-old young plants) and not of more mature plants
- Statistical analyses
- Specific corrections at indicated places throughout the text.
- Quality of the Figures
In regard to the figures´ quality, we have uploaded in the manuscript (.doc file) higher quality images and enlarged sections of the figures that were previously not legible. We have revised the size and resolution of the figures available in a .zip file as .tiff files. We would kindly suggest to the editor to make sure the reviewer can access the figures in tiff format. We have realized that the .pdf document available on the submission webpage could not have provided enough quality for a correct visualization of the figures, due to issues with format conversion. Also, in figure 3 (prev. Fig. 4), the images obtained from the KEGG Pathway tool are difficult to read when all 4 conditions are complied. We suggest, if possible, that the figure would be adapted to fit two pages to ensure it is legible. We added to the zip file the individual images for each condition.
- Use of seedlings (six-day-old young plants) and not of more mature plants
The experiments reported in this paper belong to the Project termed “Seedling Growth”, a series of three experiments performed in the International Space Station (2011-2019, including ground controls) as a result of an agreement between NASA and ESA, the American and European Space Agencies. Scientifically, the project was a synergistic cooperation between an American team led by Prof. John Z. Kiss and a European team led by Dr. F. Javier Medina, with the shared objective of understanding how gravity and light responses influence each other in plants and to better understand the cellular signaling mechanisms involved in plant tropisms--plant movement and growth.
From the very first steps of the definition of the project, our choice was to use young seedlings as experimental material, and this choice was indeed incorporated to the title of the project. The experimental setup was designed to consist of sending dry seeds to space, which would be hydrated and germinated, and seedlings grown for six days, and then collected and preserved for post-flight analysis in the ground laboratory. A highly reliable facility, the EMCS, successfully used in previous experiments, specialized for the designed operations, was selected. In fact, the EMCS was designed largely to utilize seedlings in experiments. Actually, both teams had a previous relevant experience in the use of seedlings for our respective research objectives and the work on this material had previously produced important contributions to the knowledge of the concerned biological processes. Specifically, in the case of meristematic cell growth and proliferation, seedlings were the material of choice in most studies on this topic, since the cellular and molecular events taking place in them are decisive for the establishment of the developmental pattern of the plant.
Nevertheless, we totally agree with the reviewer in the importance of performing a longer sequential study in which not only seedlings, but also adult plants could have been analyzed. This is especially important since we are dealing with a process of acclimation of plants to new environmental conditions of altered gravity, which constitute an absolutely novel challenge for plants, unknown by them throughout the history of biological evolution on the Earth. This is indeed one of the current objectives of the coming new research projects to be undertaken in a near future. However, at the time of designing the experiments reported in this paper, it was not technically feasible to propose such study (based on the limitations of the EMCS). We think that the present results are relevant per se, but they also set the basis for future experiments in line proposed by the reviewer.
- Statistical analysis
N.B. The line numbers correspond to the new version of the Manuscript document file, where all of the introduced changes are visible, as required (“All Markup” option). This also applies to the corrections listed under the heading No. 4.
L179-192 Presentation of the Statistical significance in Figure 1 (previously Fig. 2) was improved, and statistical methods are specified in the figure caption
L259 Venn diagrams of Fig. 2 (previously Fig. 3) were enlarged, to do this panels “Number of elements: specific (1) or shared by 2, 3, … lists” which are not cited in the text were erased
L385 In the Figure 4, previously figure 5, axes names were added to part A and B, and graph of part C was enlarged
L451 Venn diagrams in Fig. 5 (prev. Fig. 6) were enlarged, to do this panels “Number of elements: specific (1) or shared by 2, 3, … lists” which are not cited in the text were erased
- Specific corrections at indicated places throughout the text
To address the remaining corrections, we have introduced the following changes:
L49: “exposed” changed to “subjected”
L109 to L115: this information was already included in the Materials and Methods section and was deleted from the Results sectionL291 In the Table 1 all transcription factors were added At
L316: At5g65100 to AT5G65100
L342 Figure 3 (previous Fig. 4) was uploaded in higher resolution. If possible we would like to suggest to the editor to fit this figure in two pages due to its large size, so it can be better legible
L526-L528 explanation of anterograde and retrograde communication edited
L531 abbreviation AOX1a explained
L562 “even” was deleted (repeated)
L625 “Os” was added to the TF´s name
L644- 646 Table 3 caption was corrected (pytohormone to phytohormone) and statistical criterion was specified.
L650: “involve” substituted by “involved”
L653 to L660 The reference list was revised and information about the involvement of SA and JA in abiotic stress responses was specified. Taking into account the extensive reports on this subject relevant reviews were cited.
L677: Involvement of ABA in hydrotropism is mentioned with corresponding reference
L695: Arabidopsis thaliana to A. thaliana
L705 O2 and CO2 levels were given in the Materials and Methods section
L723-727 Figure 1 (now Fig. 6) was moved to the Materials and Methods section
L750-751: DNase treatment was specified
L787: can in the conclusions was changed to could
L770 source of KEGG Pathway tool given
Throughout the text:
- “At” was added to the TFs´ names expressed in thaliana
- All figure references were updated due to the change of previous Fig. 1 to materials and methods (now Fig. 6)
- References were adjusted to the journal format (year in bold and journal name in italics) and revised.
- All figures were imported again into the document in a better resolution
Reviewer 2 Report
The paper submitted by Herranz, Medina and colleagues reports a detailed study focusing on the interaction between red light and gravity at Mars level and microgravity as compared with 1xg. Work objects fall into the aims of the Journal and give meaningful insights into the topic. The role played by light and phototropisms as a signal for plant development in presence of low gravity is of particular interest as well as the response of photosynthesis under different gravity conditions. The methodology involves both microscopy analysis and transcriptomics. The paper is written in a friendly shape, statistics, figures and tables are appropriate to support result report and discussion. The paper can be accepted in the present form Please, check some typing error throughout the text, for example L537 “even…even”, also check the authors’ guide, while citing Figure and Table the uppercase should be used for the first letter.
Author Response
Response to the Reviewer 2
We would like to thank the reviewer for the positive analysis of our manuscript that is expressed in the reviewer’s report, as well as for the correction of specific points.
We have addressed the corrections the reviewer mentions and have revised the manuscript for correct misspelling or format issues. Concerning these, we have introduced the following changes (please notice that the line numbers correspond to the new version of the document when all of the introduced changes are visible - “All Markup” option).
L313: “Figure” was changed to “Fig.”
Table 3 caption was corrected (pytohormone to phytohormone)
L531: abbreviation AOX1a explained
L562: “Even” removed
L650: “involve” substituted with “involved”
L695: Arabidopsis thaliana to A. thaliana
L770: source of KEGG Pathway tool given
References were adjusted to the journal format (year in bold and journal name in italics) and revised.
Round 2
Reviewer 1 Report
After the revision from the manuscript entitled "From spaceflight to Mars g-levels: adaptive response of A. thaliana seedlings in a reduced gravity environment is enhanced by red light photostimulation", proposed for publication in Internacional Journal of Agricultural Sciences, all the changes and questions proposed after the revision have been succesfully solved. Consequently, I consider that this article is suitable for publication in its present form.